# Damages of Skidder and Oxen Logging to Residual Trees in Uneven-Aged Mixed Forest

Jelena Knežević , Jusuf Musić, Velid Halilović  and Admir Avdagić *

Faculty of Forestry, University of Sarajevo, 71000 Sarajevo, Bosnia and Herzegovina;
j.knezevic@sfsa.unsa.ba (J.K.); j.music@sfsa.unsa.ba (J.M.); v.halilovic@sfsa.unsa.ba (V.H.)
* Correspondence: a.avdagic@sfsa.unsa.ba

**Abstract:** The negative influence of timber harvesting on the forest environment is reflected through damage to the residual trees, regeneration, and forest soil. Considering that skidding, a popular extraction method, can cause substantial and severe damage to the remaining stand, the aim of this research was to determine damage to residual trees during skidding by an LKT 81T cable skidder, including oxen bunching. The research was conducted in eastern Bosnia and Herzegovina, in an uneven-aged mixed fir (*Abies alba* Mill.) and spruce (*Picea abies* L.) forest with pine (*Pinus sylvestris* L.) on limestone soils. Tree felling was conducted using a Husqvarna 372 XP chainsaw. Extraction operations caused damage to 6.31% of the residual trees in the stand. The most damage was "removed bark" (65.34%) and occurred on the lower parts of the tree, the butt end (55.11%) and root collar (32.39%). The average size of the damage was 197.08 cm$^2$. A statistically significant correlation was found between the damage position and the diameter at the breast height ($p < 0.05$) and the damage position and damage size ($p < 0.01$) by Spearman correlation analysis. The conducted analysis by the chi-squared test showed that there is a statistically significant difference in the proportion of damage for trees with different distances to the nearest skid road ($p = 0.0487$), but the share of damaged trees did not decrease by increasing the distance from the skid road.

**Keywords:** timber harvesting; skidder; oxen bunching; damage

## 1. Introduction

Timber harvesting has a significant influence on the forest environment. Felling and extraction operations cause damage to the residual trees, regeneration, and forest soil [1–8]. This damage leads to a lower number of trees that can potentially provide a higher quality of saw and veneer logs [9]. Among the numerous authors dealing with forest damage, there is general agreement that, due to the nature of the work, forest operations cannot be carried out without some damage to the forest ecosystem, despite the protection measures implemented. Additionally, most researchers believe that the number of mechanically damaged trees is a good indicator of the overall damage to the stand [10–13]. This damage can be determined relatively easily and accurately, and their environmental and economic consequences (decline in value, decrease in growth, and tree dieback) are better known than those caused by damage to young trees or soil [14].

Timber extraction from the felling site to landing is one of the most costly phases of forest harvesting, and regardless of which technical means are used, damage to the forest stand and soil is caused [15,16]. Investigations into the current practices of logging operations in European mountain forests have shown that tractors and skidders are used in 75%, cable yarders in 15%, and forwarders in 8% of the analyzed forest operations [17]. Forwarders are not used for timber extraction in uneven-aged forests [17]. Chainsaw and various tractors are almost exclusively used for felling and wood extraction in Bosnia and Herzegovina [18,19]. The most common tractors are skidders, entirely cable skidders [20,21], which are also the most represented in the surrounding countries [4,22,23].

Skidding, a popular extraction method, can cause substantial and severe damage to the remaining stand. The results of research have shown that most logging damage occurs during skidding operations compared to chainsaw felling operations [1,24–33]. Jourgholami [34] found that more damage occurred in the winching phase compared to the skidding phase because of the incorrect directional felling and inappropriate choice of winching routes. Damage to residual trees caused by timber skidding depends on the stand conditions, primary and secondary openness, felling method, processing method, operational preparation, and work supervision [35].

Picchio et al. [7] found that the probability of damage caused by a tractor equipped with a winch or skidder was higher on steeper slopes where larger damage and, on average, more damage per tree occurred. Behjou [36] claimed that the probability of individual tree damage decreased as the skid road cross slope decreased and the distance to the skid road edge increased. Considerable research has confirmed the thesis that the number of damaged residual trees is reduced with increasing distance from the center of the skid roads [2,34,37–39].

The share of damaged residual trees is strongly influenced by the harvest intensity and the processing method. Therefore, an increase in harvest intensity leads to an increase in the damage to residual trees caused by ground-based skidding [36]. Additionally, the cut-to-length method causes less damage to the residual stand than the tree-length method [19,40]. The level of damage to the residual stand is influenced by the direction of assortments in relation to the skid road [41] as well as the working season [42].

The damage caused by a rubber-tired skidder appears when logs scrape against stems and expose the lateral roots of the residual trees on the skid roads [43,44]. Winching rope, front and rear skidder blades, and tires can also cause damage to the residual trees [3]. Each damage to a standing tree can lead to infection by pathogenic microorganisms [45]. Infection and rot appearance are correlated with damage size and position on the tree [2]. Tavankar et al. [9] found that the wound healing rate decreased by increasing the wound width and increased the higher the wound was from the ground level.

The diameter growth of damaged trees was lower than that in undamaged trees, as well as volume increments [6,46,47].

The different technologies used for timber extraction caused different levels of damage to standing trees, soil physical characteristics, and forest regeneration [8,30,48–53]. Animal logging caused less damage to residual trees [1,54–56], regeneration [16,55,57], and topsoil [58] and less soil disturbance [59] than mechanized logging.

Considering the cited significant negative influence of forest damage, the aim of this research is to determine the damage to residual trees during skidding by an LKT 81T cable skidder, including oxen bunching. The main hypothesis of this research is that there are differences in the damage to residual trees among analyzed tree species related to the bark thickness.

## 2. Materials and Methods

The research was conducted in eastern Bosnia and Herzegovina in an uneven-aged mixed fir and spruce forest with pine on limestone soils. The study area is located between 43°57′14″ to 43°57′50″ N and 19°0′39″ to 19°1′44″ E. The elevation in the study area ranged from 1170 to 1305 m above sea level (Figure 1).

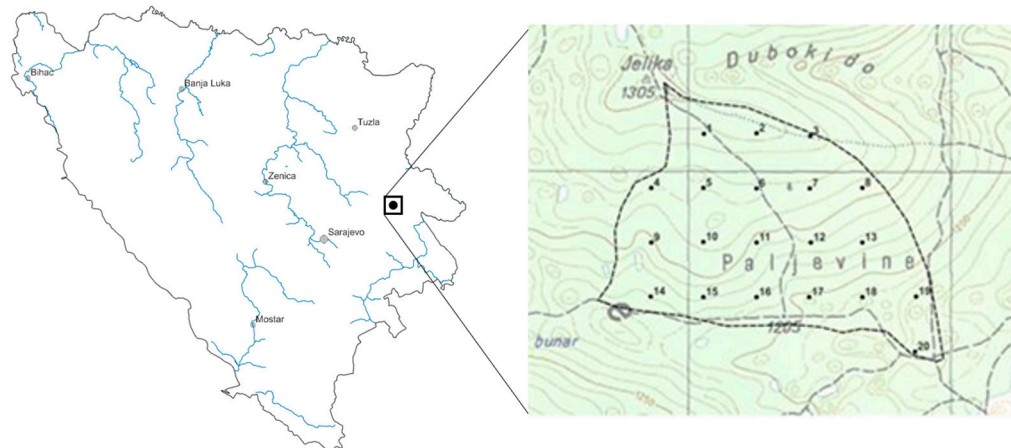

**Figure 1.** Research area with marked sample plots. 1–20: sample plots.

The average rainfall amount is 921.70 mm/m$^2$ per year, and the average slope is 5–10°
with southeastern and eastern exposure. The share of species in the mixture before felling
is as follows: silver fir (*Abies alba* Mill.), 16.26%; Norway spruce (*Picea abies* L.), 59.19%; and
Scots pine (*Pinus sylvestris* L.), 24.55%.

A group tree selection harvesting system was used. The trees for felling were marked
during 2019. The intensity of felling was 49.68 net m$^3$/ha. During August, September, Oc-
tober, and November 2020, the marked trees were felled and processed using a Husqvarna
372 XP chainsaw. The cut-to-length processing method was performed. The felling group
included a feller and assistant worker. The assortments were skidded to landing by three
LKT 81T cable skidders (Figure 2) during September, October, November, and December
2020. About half of the processed assortments (1905 m$^3$ or 50.28%) were bunched to skid
roads by animals, i.e., two oxen (Figure 3), and the remaining part by skidder's winches.

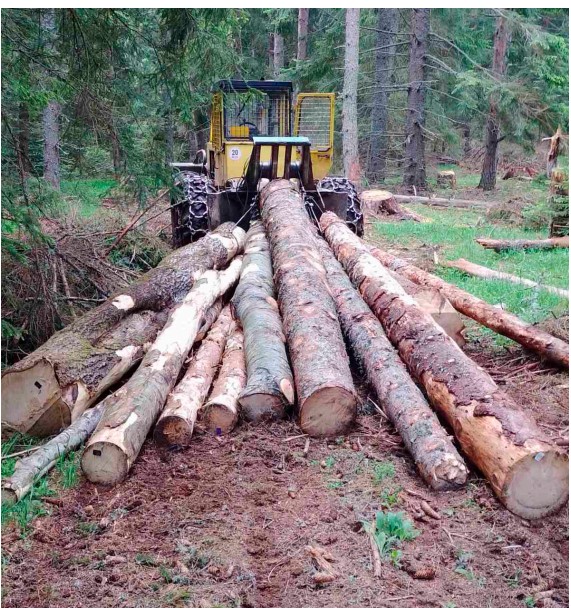

**Figure 2.** LKT 81T cable skidder.

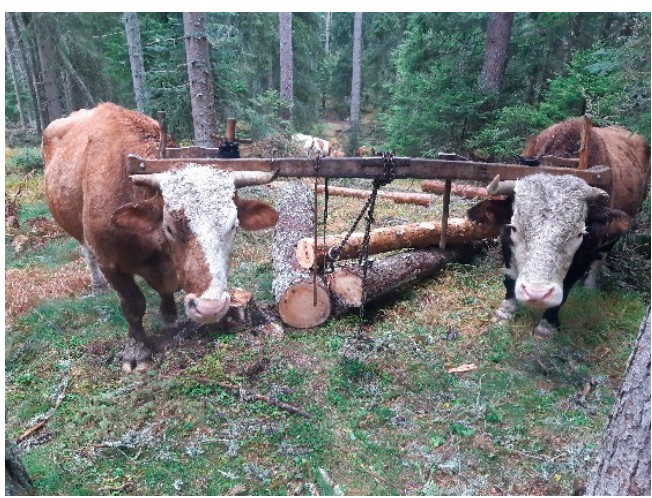

**Figure 3.** Oxen bunching.

An ox is a castrated bull trained and used for draught work. Compared to bulls, oxen are more docile (less fierce) and trainable, yet they are still strong. In forestry work, oxen are used in pairs, and their success in wood skidding depends on taming, training, and the harness used. The ox is slow but steady, hardy, strong, and easy to drive. At the end of its active life, after having been fattened, it provides a good yield in beef, so the investment in it can be recovered [60]. Compared with horse, oxen have a lower purchase price, as well as lower feeding and maintenance costs. Furthermore, oxen are calmer and secure during wood skidding. Bunching logs with oxen was carried out in work organization with one carter and a pair of oxen. The oxen were six years old with three years of work experience in the forest and weighed around 600 kg each. During the work, a short chain was used. Using a short chain between the yoke and the log is a very efficient method for skidding because the front of the log is lifted from the ground when the oxen lift their heads and move forwards. This reduces the drag resistance and the possibility of the log becoming caught on obstacles, allowing for larger loads. It also means that the log collects less dirt as it is skidded, which is important to sawmills for reducing the blunting of their saw blades [61].

Skidders were equipped with double-drum winches with a pulling force of 80 kN. The skidder group included a driver and a choker-setter. The skidder drive was limited to the constructed skid roads, which were planned and constructed during marking tree for felling. The total volume of felled and skidded wood was 3789 m$^3$ net.

Immediately after logging, the damage caused to the residual trees during the skidding operation was assessed by systematic sample plots. The grid dimension was 200 × 200 m and the sample plots were circular with a radius of 25 m and an area of 0.19625 ha (Figure 1). Systematic plot sampling consistently provides estimates similar to the results of a 100% survey for measuring residual stand damage [27,31,32,62]. The 20 sample plots were set in the analyzed compartment.

The trees on the sample plots were identified and inspected before the felling and extraction operations, and immediately after the wood extraction was completed. All damaged trees on the sample plots with a diameter at the breast height (DBH) over 5 cm were identified and inspected. The position of each damaged tree was also identified on a topographical map using the global positioning system (GPS). The following parameters were recorded for each damaged tree: tree species, diameter at the breast height (DBH), damage severity, position of damage, and size of damage.

The damage was classified as squeezed bark, removed bark, and damaged wood by damage severity [7,28,36,39], as shown in Figure 4.

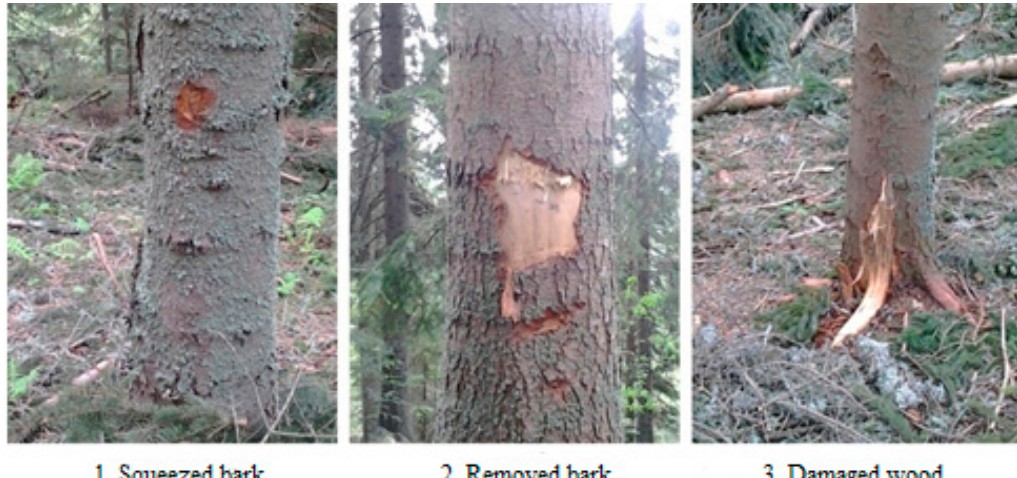

1. Squeezed bark　　　　2. Removed bark　　　　3. Damaged wood

**Figure 4.** Type of damage by severity.

The damage to trees by the damage position (Figure 5) is most commonly classified using the Meng [63] classification [5,7,8,37,39,64–66].

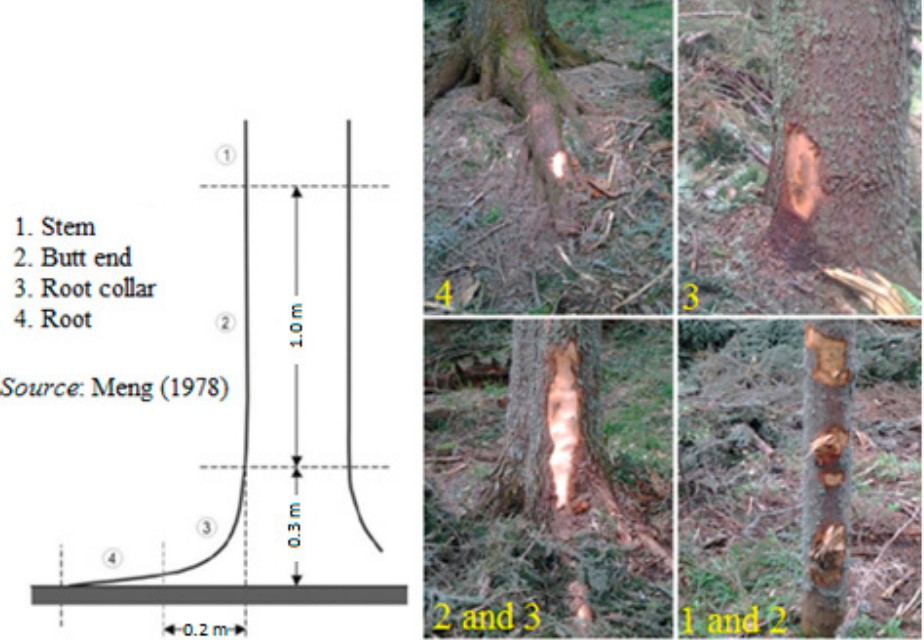

1. Stem
2. Butt end
3. Root collar
4. Root

*Source*: Meng (1978)

**Figure 5.** Classification and representation of the tree damage position [63].

The position of the damage was determined with a tape, measuring the distance between the damage center and the ground.

The maximum length and width of each damage were measured using a ruler to obtain the damage surface area using the formula for an ellipse [7,26]. The damage size was then classified by size into 4 classes: <10, 10–50, 50–200, and >200 cm$^2$ [5,27,29,37,50,64,66].

Each skidder transport cycle was recorded with a GPS built-in machine and analyzed using QGIS 3.14 with a goal to determine the number of transport cycles on each skid road. The distance of the sample plots' center to the nearest skid road was measured by measuring tape.

Statistical analyses were carried out using the Statgraphics Centurion XVI software. After checking for normality (Shapiro–Wilk W-test) and homogeneity of variance (Levene's test), the Kruskal–Wallis non-parametric multiple-comparison test was used to test the effect of the factor "diameter class" on the number of damaged trees, as well as the effect of

the factors "damage severity", "damage position", and "damage size" on the amount of damage. A Spearman rank correlation analysis was applied to test the relationship between the following: DBH vs. damage severity, DBH vs. damage position, DBH vs. damage size, damage size vs. damage severity, damage size vs. damage position, and damage position vs. damage severity. A chi-squared test was used to analyze the difference in the proportion of damaged trees among the sample plots with the different distances of the sample plot's center to the nearest skid road. The same test was used to analyze the difference in the proportion of damaged trees among the sample plots with different numbers of skidder transport cycles on the nearest skid road.

## 3. Results

The extraction operations caused damage to 89 of the 1411 residual trees on the sample plots, i.e., 6.31% (Table 1). In total, 176 damage wounds were found at a rate of 1.98 per damaged tree. The most damage was observed on *Picea abies* L. (77.53%), followed *Abies alba* Mill. (15.73%) and *Pinus sylvestris* L. (6.74%). The share of *Picea abies* L. was 30.54% higher in the structure of the damaged residual trees compared to the structure of the trees in a stand after felling (Figure 6). The share of *Pinus sylvestris* L. was 71.37% lower, while the share of *Abies alba* Mill. was slightly lower (7.85%). The share of damaged residual Abies alba L. trees (15.73%) was calculated by dividing number of damaged *Abies alba* L. trees (14) with the total number of damaged trees (89) and multiplying it by 100. The same procedure was applied for other results (Tables 2–4).

**Table 1.** Frequency of damaged residual trees.

| Tree Species | Extraction Damage | | |
|---|---|---|---|
| | Damaged Trees | | |
| | n $\pm$ SD | Trees·ha$^{-1}$ $\pm$ SD | % of Residual Trees $\pm$ SD |
| *Abies alba* Mill. | 14 $\pm$ 0.98 | 3.57 $\pm$ 0.25 | 0.99 $\pm$ 0.07 |
| *Picea abies* L. | 69 $\pm$ 2.87 | 17.58 $\pm$ 0.73 | 4.89 $\pm$ 0.20 |
| *Pinus sylvestris* L. | 6 $\pm$ 0.57 | 1.53 $\pm$ 0.15 | 0.43 $\pm$ 0.04 |
| Total | 89 $\pm$ 3.57 | 22.68 $\pm$ 0.91 | 6.31 $\pm$ 0.25 |

n: number of damaged trees on the sample plots. SD: standard deviation.

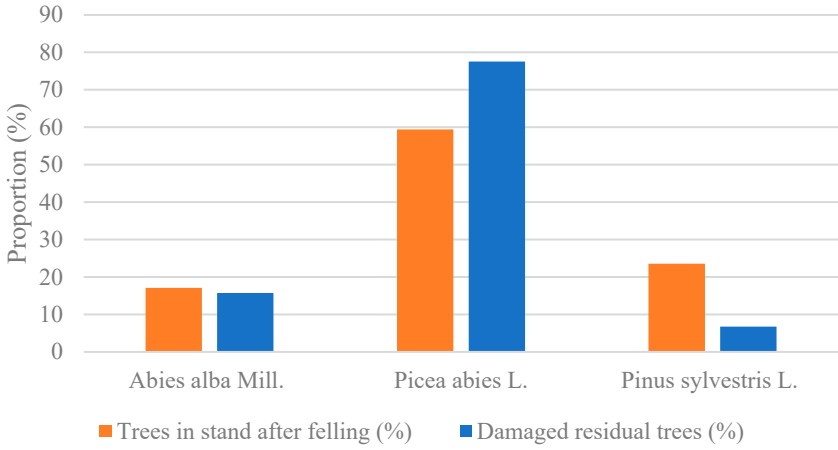

**Figure 6.** Proportion of trees in a stand after felling and damaged residual trees by tree species.

**Table 2.** Frequency of damaged residual trees by diameter class.

| Diameter Class (cm) | Damaged Trees | |
| --- | --- | --- |
| | Trees·ha$^{-1}$ ± SD | % of Residual Trees ± SD |
| 6–10 | 0.51 ± 0.08 c | 0.14 ± 0.02 |
| 11–20 | 5.61 ± 0.35 a,b | 1.56 ± 0.10 |
| 21–30 | 4.33 ± 0.31 a,b | 1.21 ± 0.09 |
| 31–50 | 9.94 ± 0.54 a | 2.76 ± 0.15 |
| 51–80 | 2.29 ± 0.15 b | 0.64 ± 0.04 |
| Kruskal–Wallis p | 0.0012 | - |
| Total | 22.68 ± 0.91 | 6.31 ± 0.25 |

Different letters show significant differences among diameter classes according to the Kruskal–Wallis test. n, number of damaged trees on the sample plots; SD, standard deviation.

**Table 3.** Frequency of damage on the damaged residual trees by damage severity, damage position, and damage size.

| Damage Severity | n ± SD | Position of Damage | n ± SD | Damage Size (cm$^2$) | n ± SD |
| --- | --- | --- | --- | --- | --- |
| Squeezed bark | 43 ± 3.18 b | Root | 17 ± 1.09 b | <10 | 4 ± 0.52 b |
| Removed bark | 115 ± 5.68 a | Root collar | 57 ± 3.10 a | 10–50 | 46 ± 2.89 a |
| Damaged wood | 18 ± 1.41 b | Butt end | 97 ± 4.13 a | 50–200 | 80 ± 3.66 a |
| Kruskal–Wallis *p* | 0.0024 | Stem | 5 ± 0.55 b | >200 | 46 ± 1.87 a |
| Total | 176 ± 7.37 | Kruskal–Wallis *p* | 0.0000 | Kruskal–Wallis *p* | 0.0001 |
| | | Total | 176 ± 7.37 | Total | 176 ± 7.37 |

Different letters show significant differences among damage severity, damage position, and damage size according to the Kruskal–Wallis test. n, amount of damage on the damaged residual trees; SD, standard deviation.

**Table 4.** Frequency of damage on the damaged residual trees by damage severity and tree species.

| Tree Species/ Damage Severity | *Abies alba* Mill. n ± SD | *Picea abies* L. n ± SD | *Pinus sylvestris* L. n ± SD |
| --- | --- | --- | --- |
| Squeezed bark | 9 ± 1.05 | 32 ± 2.30b | 2 ± 0.45 |
| Removed bark | 8 ± 0.68 | 98 ± 5.31a | 9 ± 0.94 |
| Damaged wood | 4 ± 0.41 | 14 ± 1.17b | 0 ± 0.00 |
| Kruskal–Wallis *p* | 0.6966 | 0.0038 | 0.0597 |
| Total | 21 ± 0.76 | 144 ± 3.81 | 11 ± 0.62 |

Different letters show significant differences among damage severity according to the Kruskal–Wallis test. n, amount of damage on the damaged residual trees; SD, standard deviation.

Based on the presented results (Table 1 and Figure 6), it can be concluded that the bark thickness correlates with the share of damaged trees by species. Spruce, as the species with the thinnest bark [67,68], was the most exposed to damage. Fir followed with a somewhat thicker bark than spruce [69,70], while for Scots pine, the species with the thickest bark [71], the lowest share of damage was recorded.

Significant differences were recorded for the number of damaged trees per hectare in relation to the diameter class (Table 2). The analysis of the distribution of damaged trees by diameter class (Figure 7) showed that almost half of the damaged trees belonged to the diameter class of 31–50 cm (43.82%).

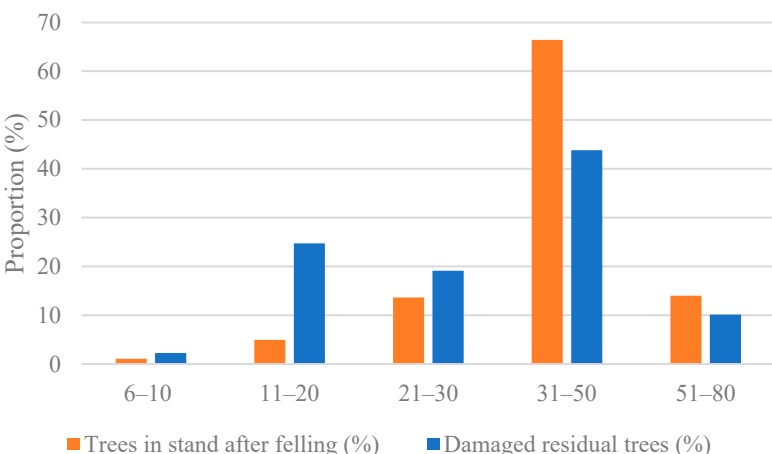

**Figure 7.** Proportion of trees in a stand after felling and damaged residual trees by diameter class.

The largest deviation between the share of the damaged residual trees and trees in a stand after felling was in the diameter class of 11–20 cm (Figure 7).

The results of the Kruskal–Wallis test showed significant differences in the amount of damage in relation to damage severity, damage position, and damage size (Table 3).

The results of the damage severity analysis indicate that the most damage was in the removed bark (65.34%), following by squeezed bark (24.43%) and damaged wood (10.23%). The most damage occurred on the butt end (55.11%). In total, 97.16% of the damage occurred on the lower parts of the trees, below 1 m from the ground, while damages to the stem (at the height above 1 m from the ground) were 2.84%.

Based on the differences in bark thickness among the analyzed tree species, an analysis of the significance of the differences in damage severity by tree species was performed and is shown in Table 4.

The results of the Kruskal–Wallis test showed significant differences in the amount of damage in relation to the damage severity only for *Picea abies* L. (Table 4). The most damage was removed bark, following by squeezed bark and damaged wood.

The average size of the damage was 197.08 cm$^2$, varying between 4.71 and 2486.88 cm$^2$ (Table 5). The analysis of the damage size showed that 2.27% of the damage was smaller than 10 cm$^2$, 26.14% was in the range of 10–50 cm$^2$, 45.45% was in the range of 50–200 cm$^2$, and 26.14% was larger than 200 cm$^2$.

**Table 5.** Review of the damage size elements.

| Damage Size (cm$^2$) | Width | | Length | | Damage Size | |
|---|---|---|---|---|---|---|
| | Range (cm) | Mean (cm) | Range (cm) | Mean (cm) | Range (cm$^2$) | Mean (cm$^2$) |
| <10 | 2–3 | 2.75 | 3–4 | 3.25 | 4.71–9.42 | 7.08 |
| 10–50 | 3–17 | 7.07 | 1–13 | 6.43 | 10.99–49.46 | 30.10 |
| 50–200 | 3–34 | 10.45 | 3–34 | 15.54 | 50.24–188.40 | 110.22 |
| >200 | 5–36 | 17.63 | 14–97 | 39.65 | 200.18–2486.88 | 531.63 |
| Total | 2–36 | 11.27 | 1–97 | 19.18 | 4.71–2486.88 | 197.08 |

The results of the Spearman correlation analysis are shown in Table 6. A statistically significant correlation was found between the damage position and DBH ($p < 0.05$) and the damage position and damage size ($p < 0.01$). In the case of the damage position and DBH, the correlation was negative because the damage on the trees with a larger DBH were closer to the ground. In the case of the damage position and damage size, the correlation was positive because the damage of a larger size was further away from the ground. The correlation between the damage size and DBH was not statistically significant ($p = 0.0703$).

**Table 6.** Results of the Spearman correlation analysis.

| Variables | N | Correlation Coefficient | *p*-Value |
|---|---|---|---|
| DBH vs. damage severity | 176 | −0.1203 | 0.1116 |
| DBH vs. damage position | 176 | −0.1575 | 0.0372 * |
| DBH vs. damage size | 176 | 0.1368 | 0.0703 |
| Damage size vs. damage severity | 176 | 0.1030 | 0.1731 |
| Damage size vs. damage position | 176 | 0.2242 | 0.0030 * |
| Damage position vs. damage severity | 176 | −0.0019 | 0.9797 |

* Statistically significant correlation.

The impacts of the distance of the sample plot's center to the nearest skid road and the number of skidder transport cycles on the nearest skid road to the proportion of damaged residual trees were analyzed by a chi-squared test (Tables 7 and 8). The conducted analysis showed that there is a statistically significant difference in the proportion of damaged trees among the sample plots with different distances of the sample plot's center to the nearest skid road ($p = 0.0487$). A statistically significant difference in the proportion of damaged trees among the sample plots with different numbers of skidder transport cycles on the nearest skid road was not confirmed ($p = 0.6911$).

**Table 7.** Analysis of the impact of the distance of the sample plot's center to the nearest skid road on the proportion of damaged trees.

| Distance of the Sample Plot Centre to the Nearest Skid Road (m) | Number of Residual Trees | Number of Damaged Residual Trees | Number of Undamaged Residual Trees |
|---|---|---|---|
| <20 | 354 | 14 | 340 |
| 21–40 | 781 | 51 | 730 |
| >40 | 276 | 24 | 252 |
| Total | 1411 | 89 | 1322 |
| Chi-squared *p* | | 0.0487 * | |

* Statistically significant difference.

**Table 8.** Analysis of the impact of the number of skidder transport cycles on nearest skid road on the proportion of damaged trees.

| Number of Skidder Transport Cycles on the Nearest Skid Road | Number of Residual Trees | Number of Damaged Residual Trees | Number of Undamaged Residual Trees |
|---|---|---|---|
| 1–5 | 191 | 14 | 177 |
| 6–10 | 275 | 19 | 256 |
| >11 | 945 | 56 | 889 |
| Total | 1411 | 89 | 1322 |
| Chi-squared *p* | | 0.6911 | |

## 4. Discussion

Oxen have traditionally been used in forestry in Bosnia and Herzegovina for more than 150 years. The art and technique of working with oxen have not been lost in many areas of Bosnia and Herzegovina, particularly in the eastern parts where oxen are still important in forestry activities, and forest enterprises continue to reach good levels of production with them. The literature on animal skidding suggests it has important ecological and economic advantages in certain conditions [72]. Pre-skidding (bunching) of processed roundwood by oxen represents the extensive way of timber harvesting that achieves benefits through: (1) reduced damage on the soil, seedlings, and remaining trees after

felling, which is important in selective forest management systems in order to protect the regeneration gaps [16,58,72,73]; (2) greater productivity of the skidder due to the previously bunched roundwood along the built skid roads [74]; and (3) absence of the need to build a dense secondary network in the context of cost reduction, but also subsequent erosion processes [75]. Therefore, skidding by oxen represents a suitable method in: (a) forests of protected nature areas, where low felling intensity is used and where there are no skid roads; (b) areas where the use of machinery is prohibited for environmental protection reasons (water protection zones); and (c) in commercial forests where the construction of skid roads is expensive or there is a high risk of erosion processes.

The extraction operations by LKT 81T including oxen bunching caused damage to 89 of the 1411 residual trees on the sample plots at a rate of 1.98 per damaged tree. The most damaged were *Picea abies* L. trees and the least damaged were the *Pinus sylvestris* L. trees. The most damage was in the removed bark (65.34%), which occurred on the butt end (55.11%). The average size of the damage was 197.08 cm$^2$. The conducted analysis test showed that there was a statistically significant difference in the proportion of damage for trees with different distances to the nearest skid road, but the share of damaged trees did not decrease by increasing the distance from the skid road.

According to the aim of this research, the share of damaged residual trees was 6.31%, which is a lower percentage compared to most previous research conducted in Bosnia and Herzegovina where the share of damaged residual trees ranged from 15.41% to 35% [33,39,76]. Similar research carried out in the same area (another forest compartment), where only oxen were used for wood skidding, determined a damage to the forest stand of only 3.32% [56], which is almost twice as low. This comparison confirms the environmental benefit of using oxen for wood skidding.

The share of damaged residual trees was within the interval of 5–15%, which is cited for tractor skidding by Martinić [14,77] in similar forests and silvicultural treatment, but mostly lower than the results of similar research when only skidders or agricultural tractors were used for timber extraction. The share of damaged residual trees found by some research was as follows: Cudzik et al. [53] in a mixed spruce and beech stand during the late thinning phase by an LKT 81T skidder (6.8 and 13.5%); Ezzati and Najafi [78] for skidding by TAF and Timberjack 450C wheel skidders in a hardwood, uneven-aged stand with shelterwood and a single-tree selection system (18.83%); Tavankar and Bodaghi [27] for skidding by a Timberjack 450C skidder in a hardwood stand with a single selective cutting (8.14%); Jourgholami [34] for skidding by a Timberjack 450C skidder in a hardwood, uneven-aged stand with single and group selective cutting methods (16.4%); Tavankar et al. [28] for skidding by a Timberjack 450 C skidder in hardwood, uneven-aged forests with the selection cutting method (11.1%); and Tsioras and Liamas [79] for skidding by agricultural tractors in beech, evenly aged stands (17–28%). Our results were also higher than the share of damaged residual trees for skidding by a LKT 81 cable skidder in mixed, uneven-aged stands (2.27% and 2.31%) and Timberjack 240C in mixed, uneven-aged stands (1.84% and 1.77%) found by Sabo and [80,81], as well as skidding by a Timberjack in hardwood, uneven-aged stands with the single and group selective cutting methods (2.3%) found by Jourgholami [40].

The results showed that the number of damaged trees·ha$^{-1}$ was 22.68, which is higher than the 15.4 and 10.4 trees·ha$^{-1}$ found by Marčeta [19] in uneven-aged beech forests during tree felling by chainsaw and skidding by an LKT 81T. However, it is important to notice that different silvicultural treatments and forest management systems were used.

The amount of damage per damaged tree was 1.98. The results of other research have shown a lower amount of damage per damaged tree: 1.04 and 1.05, 1.25 and 1.35, 1.38, and 1.33–1.90 [39,67,69,70], respectively. Jourgholami [34] found that 67% of damaged trees had 1–3 damage wounds, i.e., 70% of trees had more than 1 damage wound. Picchio et al. [7] stated that the amount of damage per tree varied between one and five, with the most common being two. The significant share of damages per damaged tree can be a consequence of the spatial position of trees in a stand or the principle of "sacrificed

tree" during extraction, which is in accordance with a small share of damaged residual trees (6.31%).

The most damage was in removed bark (65.34%). Removed bark without damage to the wood was also the most common damage according to Gurda et al. [39], with a share of 67% and 92% for uphill and downhill winching with a Timberjack 240 C skidder, respectively. Dudáková et al. [8] found that the most common type of damage was wood exposed but undamaged (61.3%), which is compatible with the removed bark from this research. Additionally, Picchio et al. [7] stated that the most common damage during extraction by tractor with a winch or cable skidder was the bark removed, reaching up to 70% in pine stands and 30–40% in beech stands.

The most damage occurred on the butt end at a height of 0.3–1 m from the ground (55.11%). The results of research on damage to residual trees during logging by chainsaw and LKT 81T skidder also showed the most damage was to the butt end [33]. Gurda et al. [39] also found that damage to the butt end was the most common, with a share of 53% during uphill winching by a Timberjack 240C skidder. Damage to the butt end was the most common according to Tsioras and Liamas [79], with a share of 40%. Dudáková et al. [8] and Cudzik et al. [53] found that damage to the butt end was not the most common damage but had a share of 29% and 41% in all damage. In total, 97.16% of the damage occurred on the lower parts of the trees, below 1 m from the ground, which is similar to the 97%, 80%, 80–94%, 82%, 80.77%, 97%, 78%, 98.6%, and 90.3% found by Jourgholami [40], Tavankar et al. [28], Tsioras and Liamas [79], Tavankar et al. [6], Milou and Diminos [29], Cudzik et al. [53], Tavankar et al. [82], Tavankar et al. [47], and Dudáková et al. [8], respectively, for logging with a chainsaw and a skidder/tractor with winch.

The average size of the damage was 197.08 cm$^2$, which is smaller than the 610 cm$^2$ found by Tsioras and Liamas [79], 608 cm$^2$ found by Sabo [81], 460.35 and 262.40 cm$^2$ found by Nikooy et al. [25], and 228.29 cm$^2$ found by Sabo [80]; and higher than the 65.16 and 81.66 cm$^2$ found by Gurda et al. [39], 92.26 cm$^2$ found by Halilović et al. [33], 145.32 cm$^2$ found by Dudáková et al. [8], 165.7 cm$^2$ found by Tavankar and Bonyad [31], and 145.4 and 176.6 cm$^2$ found by Danilović et al. [41]. The most common was damage ranging in sizes of 50–200 cm$^2$ (45.45%), which was confirmed by Danilović et al. [41] and Tsioras and Liamas [79] for one site with a share of 40.4%, while on the other three sites, damage of sizes >200 cm$^2$ were the most common. Picchio et al. [7] and Halilović et al. [33] found that the most common was damage ranging in sizes of 25–100 cm$^2$; Tavankar et al. [6] found it to be 101–200 cm$^2$, and Tavankar and Bodaghi [27] found <10 cm$^2$. The range of damage size (4.71–2486.88 cm$^2$) was similar to 4–2000 cm$^2$ as found by Sabo [80], smaller than 28–5250 cm$^2$ as found by Sabo [81], and higher than 12–600 cm$^2$ as found by Danilović et al. [41].

A statistically significant correlation was found between the damage position and DBH and the damage position and damage size. Tavankar et al. [6] analyzed the correlation between the same variables and found that the only statistically significant correlation was between the damage severity and damage position. In the case of the damage position and damage size, the correlation was positive because the damage size increased with increasing damage height from the ground, which was confirmed by Tavankar et al. [82]. The correlation between the damage size and DBH was not statistically significant, and a further regression analysis was not performed.

The conducted chi-squared test showed that there was a statistically significant difference in the proportion of damaged trees among sample plots with different distances of the sample plot's center to the nearest skid road. It was found that the share of damaged trees did not decrease by increasing the distance from the skid road. A statistically significant difference in the proportion of damaged trees among the sample plots with different numbers of skidder transport cycles on the nearest skid road was not confirmed.

Damages to residual trees can be additionally reduced by organizational measures and appropriate planning of the felling site with visible marks on terrain, and also by the preventive protection of the most threatened trees by setting the "protectors". However,

adequate control of forest harvesting activities, which is often absent, is the most important for high-quality work in forests.

The main hypothesis of this research has been confirmed. Spruce, as the species with the thinnest bark, was the most exposed to damage. Fir had a somewhat thicker bark than spruce, while for Scots pine, as the species with the thickest bark, the lowest share of damage was recorded.

This is the first research of damage to residual trees caused by LKT 81T skidder, including oxen bunching. This is the most commonly used technologies in forestry practice for timber extraction in Bosnia and Herzegovina when animals (horses or oxen) are included. The conducted research did not include the other negative aspects of analyzed timber extraction, such as damage to regeneration and damage to forest soil. Future researchers of timber extraction by skidder and animals should include all negative effects to residual trees, regeneration, and forest soil.

## 5. Conclusions

Mechanical damage to trees has multiple negative impacts, which are reflected in the reduction in the vitality, productivity, and value of trees as well as other components of the forest ecosystem. Unfortunately, this very important and complex problem in BiH's forestry practice and science has not received adequate attention. One of the ways to reduce these negative impacts is the use of environmentally friendly technologies of wood skidding, e.g., the use of oxen for the bunching of wood. The results of this work show that the share of damaged residual trees during wood extraction by a skidder, including oxen bunching, was much lower compared to that during the winching of wood with rope.

**Author Contributions:** Conceptualization, J.K. and V.H.; methodology, J.K. and J.M.; software, A.A.; validation, J.K., J.M., and V.H.; formal analysis, A.A. and J.K.; investigation, J.K., J.M., V.H., and A.A.; data curation, J.K.; writing—original draft preparation, J.K. and J.M.; writing—review and editing, V.H. and J.K.; supervision, J.K. All authors have read and agreed to the published version of the manuscript.

**Funding:** This research received no external funding.

**Data Availability Statement:** Raw data were generated at the Faculty of Forestry, University of Sarajevo, Zagrebačka 20, 71000 Sarajevo. Derived data supporting the findings of this study are available from the main author, J.K., or the corresponding author, A.A., upon request.

**Conflicts of Interest:** The authors declare no conflict of interest.

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
