# Peer review of "Damages of Skidder and Oxen Logging to Residual Trees in Uneven-Aged Mixed Forest"

_forests, doi:10.3390/f14050927_

Round 1

Reviewer 1 Report

Authors presented an interesting and very well written manuscript. In my opinion there are just minor revisions needed (listed):

- Lines 38 – 40: Please support with a reference.

- Line 121 (Figure 3): Please add numbers to figures also.

- Line 151 – 152: Please check the statement.

I suggest describing the reasons for using oxen bunching in more detail (in relation to density of skid trails, etc.)  

Author Response

Dear Reviewer,

Thank You for taking the time to review our paper. We are pleased to hear that You found our manuscript to be very well written. We have made changes according to Your suggestions. The reference was added in lines 38-40. In line 121 (Figure 3) we added numbers to the figures. We corrected the mistake in lines 151-152. Word „even“ was changed by the word „slightly“. The reasons for using oxen bunching, as well as, differences in oxen and horse bunching were written in „Material and methods“ and „Discussion“.

Best regards,

Authors

Reviewer 2 Report

Good research, it was a pleasure to read it.

The only thing missing in the manuscript of the paper submitted for review are the recommendations (two to three sentences) of the application of the conducted research in the end of conclusions.

Namely, the pre-skidding – buncing of processed roundwood by oxen represents extensive way of timber harvesting, which achieves benefits through: 1) reduced damage to the remaining trees after felling, which is important in selective forest management in order to protect the regeneration gaps, 2) greater productivity of the skidder due to the previously bunched roundwood along the built skid  roads, 3 ) absence of the need to build a dense secondary network in the karst area in the context of cost reduction, but also subsequent erosion processes.

With the above, the authors should point out the area of application of the conducted research: 1) commercial forests or 2) forests of protected nature areas or 3) forests in the area of the ecological network – NATURA 2000 with a short explanation.

In addition, I noticed a couple of „small things“ in the text that will further improve the submitted manuscript for review:

1)      line 36 the words „Transport logs“ should be replaced by the words „Timber extraction“

2)      lines 62 and 63 the words „short-log“ should be replaced by the words „cut-to-length“

3)      line 63 the words „long-log“ should be replaced by the words „tree- length“

4)      line 68 the word „rare“ should be replaced by the word „rear“

5)      line 99 the words „assistant worker“ should be replaced by the words „choker-setter“

6)      line 100 the words „skid trails“ should be replaced by the words „skid roads“

7)      line 130 the words „skid trail“ should be replaced by the words „skid road“

8)      line 216 the word „assortment“ should be replaced by the words „cut-to-length“

9)      line 224 the word „unevenlt“ should be replaced by the word „uneven“

Author Response

Dear Reviewer,

Thank You for taking the time to review our paper. We appreciate Your suggestions for improving conclusions by adding the recommendations of the application of the conducted research. We have made changes according to Your suggestions. The words „Transporting logs“ were replaced by the words „Timber extraction“ in line 36. The words „short-log“ were replaced by the words „cut-to-length“ in lines 62 and 63. The words „long-log“ were replaced by the words „tree- length“ in line 63. The word „rare“ were replaced by the word „rear“ in line 68. The words „assistant worker“ were replaced by the words „choker-setter“ in line 99. The words „skid trails“ were replaced by the words „skid roads“ in lines 100 and 130, as well as the entire text. The sentence in line 216 was changed. The word „unevenlt“ was replaced by the word „unevenly“ in line 224.

Best regards,

Authors

Reviewer 3 Report

The authors have done a good job, the study is novel and interesting enough to deserve publication. However, there are several improvements that must be made :

-The abstract focuses on sample plots. Please focus you wording rather on the stand itself. The sampling should have been made correctly enough so that you can write about the stand as a whole, rather than just reporting the results of the sampling. The results of your specific sampling is of no interest to international readers.

-The Title includes oxen but very little of them are mentioned in the M&M and results. Please describe more (or provide reference to) how the oxen bunching was performed. Also, just as importantly, half of the bunching was done by oxens and half by winching. Were there any differences in damage, productivity, etc. between these two methods? This question comes to mind when reading you manuscript, and you must answer that question somewhere in your results. 

-Please, for international readers, provide a map of the study area.

-The Results section is ok but could be vastly improved by using the text to guide the reader to the important tidbits of each results table. Eg. Table 3, what is the reader supposed to pay attention to in it (ie. what is the most interesting fact about this table?). Please provide such a fact/guidance for each results table and figure.

-The Discussion section must be improved. At present, the Discussion section is very narrowly focused and lacks several important parts. A Discussion section should contain the following parts (parts 1, 3, 4 , and 5 are presently lacking): 

1. Summarize your key finding in the first sentences

2. Compare (your findings) with prior knowledge

3. Interpret your findings and give explanations to why the findings are that way

4. Strengths and Limitations of your study

5. Make recommendations (practical implementation/subsequent studies)

6. Summary/conclusion 

-The conclusions are broad enough and well-written enough to be of interest to international readers. Good job there.

-There are many wordy/long sentences that impede reader understanding of the messages that you want to convey. Eg. Line 287-290, Line 141-144. There are also som some run-on sentences e.g. Line 212-216. Please rewrite all of such sentences (there are more) that are long and hard-to-understand. 

Author Response

Dear Reviewer,

Thank You for taking the time to review our paper. We are pleased to hear that You found our study to be novel and interesting. We appreciate Your suggestions for improving abstract, and we have made changes according to Your suggestions. Damage caused by extraction operation was related to analyzed stand, and not to sample plots. The last sentence in abstract was changed and written generally, not relying to sample plots. We described used animals (oxen) and work technology more precisely in „Materials and Methods“ section. Also, we compare damage to residual stand determined in this research during oxen bunching/skidder winching+LKT 81T skidding and the damage to residual stand during oxen skidding in similar stand and terrain conditions. The map of study area with sample plots was inserted. The most important results of each table or figure were already listed in text before or after table/figure. The „Discussion“ section was improved according to Your suggestions. Thank You for useful and detailed comments. We changed sentences in lines 287-290, 141-144 and 212-216.

Best regards,

Authors

Reviewer 4 Report

Unfortunately the publication title promises more than it implies. The authors described one type of forest operation, without comparing it with another operation (even the simplest combination of a skidder with an ox and without an ox). This means that this work cannot be classified as a scientific publication, perhaps as a scientific report.

However, I would like to point out that the statistical analyzes used  and analysis of the literature in the field of forestry operations with a skider show the scientific maturity of the authors to create the publication.  

In the opinion of the reviewer, in order for it to be a scientific publication, this forest operation (Skider + ox) should be compared with another forest operation or in other conditions (older forest or younger forest like: early thinning, late thinning, final pruning). If there is no such possibility and the authors have only such research material as the one presented, I suggest:

1/ change of the title taking into account the influence of the tree species from the main stand on the size and frequency of damage affected by skider and ox;

2/ change of the purpose of the work - verification of the sensitivity of the species and the size of the resulting damage in the forestry operation (skider + ox);

3/ creating a working hypothesis (maybe something in the direction of the thickness of the bark of the three analyzed tree species and the structure of their lower part of the trunk (root influxes, root neck) ...;

4/ change in statistical analyzes towards comparing tree species: fir / spruce / pine. 

I assume that finally it will be possible to show statistically significant differences for tree species (of course, taking into account the different sizes of the three populations of species) and indicate in which stands such a forest operation can be recommended.

Author Response

Dear Reviewer,

Thank You for all Your comments. The forest of Bosnia and Hezegovina (BiH) are dominantly mixed unevenly agged (Second National Forest Inventory in BiH, 2006-2009), mainly managed by the group selection management system. The dominant tree species are: Fagus sylvatica L., Abies alba Mill. and Picea abies L.. Felling of trees with diameter from 5 cm to 80 cm is performed in periods of ten years according to applied management system. Final felling and thinning are performed simultaneously, in the same forest compartment.  There is no possibility to compare damage to residual stand between older and younger forests, and among early thinning, late thinning and final prunning. We compare damage to residual stand determined in this research during oxen bunching/skidder winching+LKT 81T skidding and the damage to residual stand during oxen skidding in similar stand and terrain conditions. The tittle was changed and now is: „Damages of Skidder and Oxen Logging to Residual Trees in Uneven-Aged Mixed Forest“. Damage to certain tree species was analyzed taking into consideration bark thickness (text bellow figure 7). Also, the Kruskal–Wallis test was used to test the effect of the factor „damage severity“ on the amount of damage by tree species (table 4). The test showed significant differences in the amount of damage in relation to damage severity for Picea abies L. The most damage was removed bark, following by squeezed bark and damaged wood.

Best regards,

Authors

Round 2

Reviewer 3 Report

The authors have improved their manuscript. Unfortunately, there are several improvements that the authors have not made, and which they must make before the manuscript can be published:

-As another reviewer and myself pointed out, half of the bunching was done by oxens and half by winching. Were there any differences in damage, productivity, etc. between these two methods? This question comes to mind when reading you manuscript, and you must answer that question somewhere in your results. 

-The Results section is ok but should be improved by using the text to guide the reader to the important tidbits of each results table. Eg. Table 3, what is the reader supposed to pay attention to in it (ie. what is the most interesting fact about this table?). Please provide such a fact/guidance for each results table and figure.

-The Discussion section must be improved. At present, the Discussion section is very narrowly focused and lacks several important parts. A Discussion section should contain the following parts (parts 1, 4 , and 5 are still lacking): 

1. Summarize your key finding in the first sentences

2. Compare (your findings) with prior knowledge

3. Interpret your findings and give explanations to why the findings are that way

4. Strengths and Limitations of your study

5. Make recommendations (practical implementation/subsequent studies)

6. Summary/conclusion 

Author Response

Dear Reviewer,

Thank You for taking the time to review our paper. Timber extraction in analysed forest compartment was conducted by simultaneous use of oxen and skidder winch for wood bunching/winching. Different types of wood bunching/winching were not spatially separated within the forest compartment in order to analyze the differences in the damage of the stand. Respectively, oxen bunched wood which was at a distance greater than the length of the winch rope to skid roads. Skidder winched wood which was at a distance smaller than the length of the winch rope to skid roads. According to Your suggestions, we added the comparative view of stand damage during oxen bunching/skidder winching+LKT 81T skidding and oxen skidding in similar stand and terrain conditions. The damage to stand during oxen skidding was smaller (3.32%) compared to results of this research (6.31%). Skidder productivity with different types of winching/bunching (winch or oxen) was not the aim of this research. Unfortunately, we are not able to provide data on skidder productivity in the case of wood bunching by oxen, because that there are no published results in Bosnia and Herzegovina and other countries with similar stand and terrain conditions. We describe the way of calculating the share of damaged trees (%) in text before Table 1. In Discussion section, parts 1, 4 and 5 were added.

Best regards,

Authors

Reviewer 4 Report

Before describing the purpose of the work, it is necessary to indicate in the background what was the working hypothesis that prompted the authors to carry out this research and what exactly it contained!

My suggestion - to accept the hypothesis related to the influence of the frequency and size of damage on the species and thickness of the bark. They felt that the authors assumed that the damage would be the same or different for the species when using this type of skidding.

The scientific purpose of the work has not changed! The aim of the work was, as the reviewer understands, a quantitative and qualitative analysis of damage to trees caused by combined skidding (skider + ox), taking into account species with different bark thickness (fir, spruce, pine). 

Line: 376-379.  These two sentences are not a conclusion at all - it's part of the discussion - please move it to the previous chapter.

Conclusions in a scientific publication must correspond to the stated purpose of the research. Please keep this in mind and correct this paragraph accordingly.

Author Response

Dear Reviewer,

Thank You for all Your comments. We wrote the hypothesis according to Your suggestions: „The main hypothesis of this research is there are differences in damage to residual trees among analyzed tree species, related to the bark thickness.”. Two sentences from Conclusions section (lines 376-379) were moved to Discussion section.

Best regards,

Authors